# A Fifteen-Year Analysis of Rare Isolated Fallopian Tube Torsions in Adolescent Children: A Case Series

**DOI:** 10.3390/diagnostics9030110

**Published:** 2019-09-04

**Authors:** Cengiz Güney, Abuzer Coskun

**Affiliations:** 1Cumhuriyet University Medical Faculty, Department of Pediatric Surgery, 58140 Sivas, Turkey; 2Sivas Numune Hospital, Department of Emergency, 58030 Sivas, Turkey

**Keywords:** emergency department, acute abdominal pain, isolated tubal torsion, detorsion, salpingectomy

## Abstract

Isolated tubal torsions presenting to the emergency department are a very rare cause of pediatric acute abdominal pain. Since making the diagnosis early is of importance in terms of affecting tubal damage and fertility, we aimed to evaluate cases of isolated tubal torsions in light of the literature. This study included 10 patients under 18 years of age who presented to the emergency department with abdominal pain between January 2003 and December 2018. The mean age was 14.5 ± 1.43 years (range: 12–17 years). The demographic characteristics, surgical findings and techniques, and concomitant pathology results of these patients were retrospectively evaluated. The reason for presenting to the emergency department for the 10 patients included in the study was abdominal pain. The mean duration of hospital admission with pain was 4.97 days. The onset of pain was less than 24 h in seven patients (70%) and more than 24 h in three patients (30%). Of the patients, nine (90%) had tenderness in the lower abdominal quadrant, five (5%) had defense, and three (30%) had rebound. Nausea, vomiting and leukocytosis were present in 50% of the cases. Right and left tubal involvement of the cases was equal. Seven (70%) of the isolated tubal torsions were accompanied by paraovarian cysts. Eight patients (80%) underwent open surgery and two (20%) underwent laparoscopic intervention. Detorsion was performed on five (50%) patients and salpingectomy was performed on five (50%) patients. Isolated tubal torsion should be considered in children presenting with acute abdominal pain in early adolescence. Early diagnosis is important for the preservation of fertility.

## 1. Introduction

Isolated fallopian tube torsion (IFTT) is the rotation of the fallopian tube on its own axis without ovarian torsion. The incidence is estimated to be 1/1,500,000 women [1]. The etiology is uncertain. Anatomical abnormalities such as long mesosalpinx, hydro/hematosalpinx, tubal mass/neoplasm, adnexal mass (ovarian/paraovarian tumor), and physiological abnormalities such as abnormal peristalsis or periovulatory spasm, hemodynamic abnormalities such as venous congestion, the Sellheim theory indicating sudden body position changes, trauma, pelvic inflammatory disease, pelvic adhesion, previous surgery/disease such as tubal ligation, and enlarged uterus/uterine mass may play a role in the etiology [2,3]. Tubal torsion occurs more frequently on the right side. Abdominal pain is present in all cases. Nausea and vomiting may be variable. Early diagnosis and treatment are important in terms of fertility preservation. No specific symptoms, clinical findings, imaging or laboratory characteristics have been determined so far [4]. Diagnosis of isolated fallopian tube torsion in the US is rarely made during the adolescent period [5]. It is rarely diagnosed before surgery. Detorsion or salpingectomy is performed as the treatment.

The aim of this study was to present the results of 10 patients with IFTT in light of the literature and to contribute to the diagnosis and treatment.

## 2. Materials and Methods

### Study Design and Population

Patients diagnosed with isolated fallopian tube torsion between January 2003 and December 2018 were retrospectively analyzed using the electronic medical record system. The study included girls aged between 1 month–18 years and patients diagnosed with isolated fallopian tube torsion intraoperatively. Patients under 1 month of age and above 18 years of age, and patients who had fallopian tube torsion with ovarian torsion were excluded from the study. The patients’ age, the characteristic and localization of abdominal pain, presence of nausea and vomiting, physical examination findings, preoperative laboratory and radiological imaging findings, affected side, surgical findings, technique and procedure, presence of concomitant pathology, and examination results were recorded.

## 3. Statistical Analysis

The data obtained from this study were collected from the hospital’s electronic medical record system, and then analyzed by the SPSS 15.0 software package. The descriptive statistics were expressed as mean ± standard deviation or median (minimum–maximum) for continuous variables and as number of cases and percentage (%) for nominal variables. Other data were analyzed using Microsoft Excel and simple descriptive statistics.

## 4. Results

The mean age of the 10 patients included in the study was 14.5 ± 1.43 (range: 12–17) years. The reason for presenting to the emergency department was abdominal pain. The mean duration between the onset of pain and hospital admission was 4.97 days (6 h–1 month). Of the patients, five had abdominal pain in the left lower quadrant, two in the right lower quadrant, one in the lower quadrant, and two had abdominal pain in the whole abdomen. All sexually inactive patients, except one, were menstruating. Of the patients, five complained of nausea and three had concomitant vomiting. Two of the patients had nausea and non-bilious vomiting. The patients had no complaints of trauma, constipation, diarrhea, vaginal discharge, or dysmenorrhea. Only one patient had dysuria. Of the patients, 90% (nine) had lower quadrant tenderness, five had defense, and three had rebound on physical examination (Table 1).

Five patients had a high white blood cell (WBC) level (>11,000/mm^3^), while five patients’ WBC levels were within the normal range (4000–11,000/mm^3^). The mean time to onset of symptoms was 8.4 days in patients with a high WBC value, while it was shorter in those with a normal WBC value at 0.95 days. Neutrophilia was detected in 70% of the patients. In the patients with relatively low lymphocyte percentage, platelet counts were within the normal range.

The urea, creatinine, amylase, aspartate transaminase, alanine aminotransferase, gamma-glutamyl transferase, sodium, potassium and C-reactive protein (CRP) values of all patients were normal. The values of three patients whose alpha-fetoprotein (α-FP), β-human chorionic gonadotropin (β-HCG), and cancer antigen-125 (CA-125) were studied were normal.

All patients underwent direct abdominal X-ray and Doppler ultrasonography (USG) and two patients underwent magnetic resonance imaging (MRI). The direct abdominal X-rays of all patients were evaluated to be normal. A paraovarian cyst/mass was detected in seven patients who had undergone ultrasonography, while two patients were evaluated to be normal. Because of the presence of free fluid in the pelvic region of one patient, it was considered to be an ovarian cyst. Four patients received the pre-diagnosis of ovarian torsion due to the absence or reduction of ovarian blood supply. Pelvic free fluid was identified in four patients and a paraovarian cyst was detected in five patients. MRI was performed on one appendectomized patient with the diagnosis of a paraovarian mass and this patient was preoperatively diagnosed with IFTT (Figure 1).

Nine of the patients underwent surgery on the day of admission. One patient was operated on four days later after further examinations were performed due to non-relieving pain, previous abdominal pain attacks and the detection of a paraovarian solid mass on USG. Two patients were operated on using the laparoscopic technique, while eight underwent open surgery. Torsioned tubes were equal on both sides. Five patients did not have any additional intervention because of normalized blood supply after detorsion, while the other five patients underwent a cystectomy/salpingectomy (Figure 2). The pathological examination of the resected material revealed changes such as chronic salpingitis along with serosal cysts (Table 2).

## 5. Discussion

Isolated fallopian tube torsion (IFTT) is one of the rare causes of acute abdominal pain in adolescent girls. In the literature, it is usually presented as case reports. In recent years, there are publications in the form of case series [6,7,8]. Tubal torsion more commonly arises on the right side. The reason for this is that the mobility of the left tuba with the sigmoid colon is more restricted and right lower quadrant pains are diagnosed by opening the abdomen with suspicion of acute appendicitis [9,10]. In our study, it was present on both sides with the same frequency. It was also similar to the results of a multicenter study by Bertozzi et al. which found an equal number on both sides [9].

Paraovarian cysts/masses have been accused in the etiology of IFFT. A paraovarian cyst was detected in five of our cases. In addition, hydro/hematosalpinx was identified in one patient. There are studies that associate isolated fallopian tube torsion with some sports involving sudden body movements [11,12]. In our study, two patients were actively engaged in sports.

Lower abdominal pain is common in IFTT; however, lower abdominal pain is not specific to IFTT, and similar manifestations may also be seen in other surgical pathologies causing acute abdomen [3]. All patients had abdominal pain complaint. Eight patients had abdominal pain in the lower abdominal region, while two had abdominal pain in the whole abdomen. The onset of pain ranged from six hours to one month from admission and was similar to the study by Harmon et al. [13]. The complaint of nausea and vomiting varied in the patients.

On physical examination, there was no specific examination finding suggestive of IFTT. Nine patients had tenderness, five had defense, and only three had rebound. Five patients had a WBC count above the normal range, while the other patients’ WBC counts were within the normal range; however, seven patients had neutrophilia. The presence of neutrophilia in 70% of the patients was remarkable. CRP was within the normal range in all of our patients. Our patients had no specific laboratory findings of IFTT, and as stated by Bertozzi et al. [7]., the contribution of laboratory parameters to the diagnosis was not clearly obvious in IFTT patients.

It is difficult to diagnose isolated fallopian tube torsion preoperatively since there is no pathognomonic imaging method, specific symptom or characteristic laboratory finding [14]. The diagnosis is usually made during surgical intervention. In our series, one patient was preoperatively diagnosed, while other patients were diagnosed during surgery. The MRI of the patient diagnosed preoperatively revealed thickened fallopian tube walls, tubal rotation and hydro/hematosalpinx, which were evaluated in favor of IFTT. Since there are no specific findings by US, the rate of diagnosis of fallopian tube torsion is low [15]. Cases are usually diagnosed with ovarian torsion, adnexal cyst or acute appendicitis. The preoperative diagnosis of four patients was ovarian torsion. Five patients were diagnosed with a paraovarian cyst on USG. IFTT was intraoperatively identified on the side where these cysts were present and torsioned fallopian tubes were removed when evaluated to be necrotic. Paratubal cysts, called hydatid cysts of Morgagni, are very rarely neoplastic [16]. In the study, the cystic structures that five patients had were diagnosed as a simple serosal cyst as a result of pathological examination. In our patient preoperatively diagnosed with a solid mass, only a cystic structure was found during the operation.

Pathologies such as appendicitis, ovarian torsion, ectopic pregnancy, pelvic inflammatory disease, and ruptured ovarian cysts should be taken into consideration in the differential diagnosis [10]. In terms of preserving fertility, early diagnosis and early surgical intervention may be possible by preserving the fallopian tube and avoiding possible salpingectomy. Saving the fallopian tube is a rare condition due to the difficulty in early diagnosis. Mazouni et al. emphasized that a duration of longer than 10 h from the onset of pain to surgery increases the risk of tubal necrosis [17]. In our study, half of the patients did not have any tissue loss because of the normalization of blood supply to the fallopian tube after detorsion. Of these patients, three were operated on within the first 24 h from the onset of their complaint, while the other two were operated more than 24 h later.

Salpingectomy or detorsion is the preferred method of surgical treatment. As laparoscopic surgery has become widespread in recent years, it is possible to both confirm the diagnosis and to treat the patients. With the widespread use of single port laparoscopic surgery in recent years, good cosmetic results will also be obtained [18]. Laparoscopic surgery was performed on only two of our patients. This is due to the late onset of laparoscopy in our clinic. The necrotic appearance of the tubes is taken as a reference when deciding on salpingectomy; however, in recent studies, the observation of healthy cilia cells in the pathological examination of the removed tubes leads to a discussion of maybe leaving the fallopian tubes in place as in ovarian torsion [19]. However, in such a case, it is recommended to follow up patients with USG until adulthood. However, no matter how necrotic it appears, the conservative removal of a fallopian tube does not affect morbidity. This reveals the need for future studies in terms of the functionality of fallopian tubes.

Isolated fallopian tube torsion was equally visible on both sides. No specific examination findings or US findings were found in the patients. Surgical intervention and timing were an important parameter in the diagnosis, treatment and prevention of loss of tissue.

The most important limitation of the study is the retrospective and single-center design. Another important limitation is the small sample size and the selection of very rare cases. Moreover, the difficulty in accessing the records and radiological imaging and the inability to obtain adequate information on the medications used and personal history of the patients were the main limitations.

## 6. Conclusions

In this study, no specific clinical, physical examination or laboratory findings were identified when the results of 10 patients with IFTT were analyzed over a period of 15 years. IFTT is a cause of acute abdomen which should be kept in mind in patients presenting with lower abdominal quadrant pain. Although USG is not very helpful in the diagnosis, the presence of paraovarian cystic masses in the affected area is an important parameter in terms of diagnosing IFTT. Preservation of fallopian tubes in terms of fertilization is possible with early surgical intervention. Although there are competing views regarding the preservation of necrotic fallopian tubes in recent times, there is not sufficient information about the long-term results. Further studies are needed to be conducted in this respect.

## Figures and Tables

**Figure 1 diagnostics-09-00110-f001:**
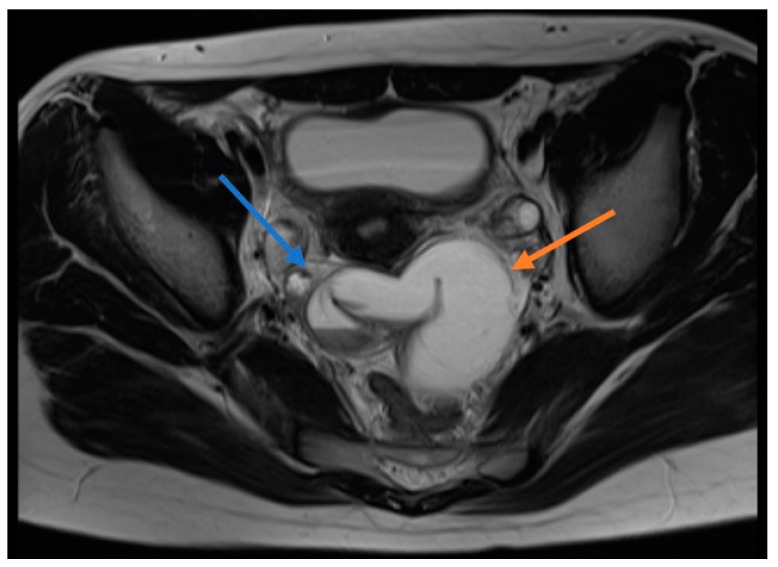
Magnetic resonance image (MRI) of a patient diagnosed with preoperative isolated fallopian tube torsion. T1A sections show blue arrow torsion, red arrow indicates hydrohematosalpinx.

**Figure 2 diagnostics-09-00110-f002:**
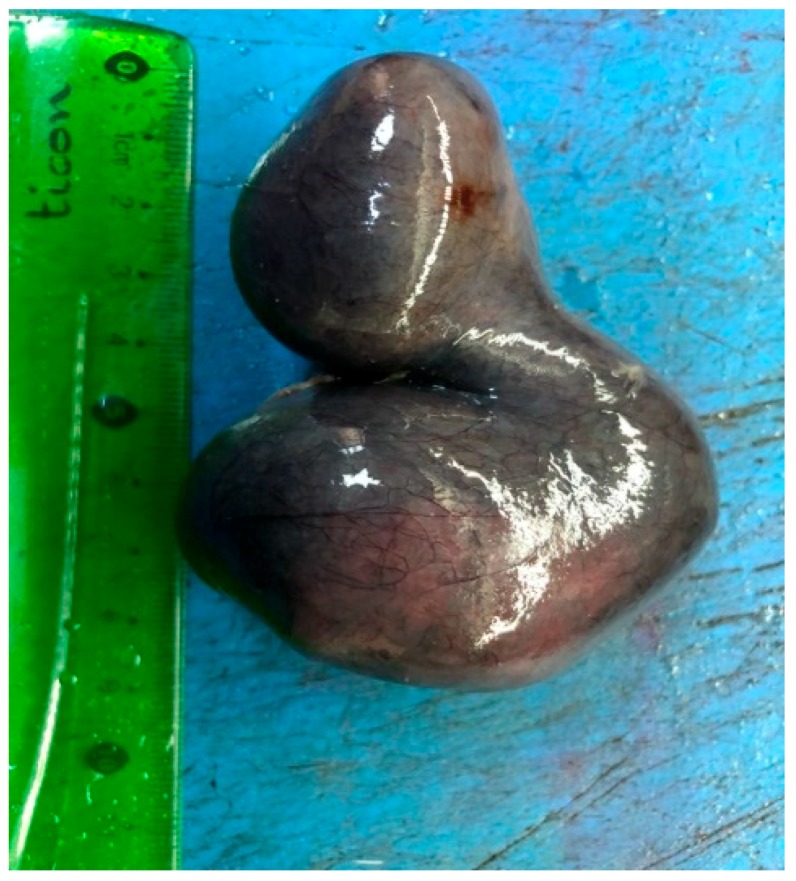
Isolated fallopian tube torsion material caused by hydrohematosalpinx.

**Table 1 diagnostics-09-00110-t001:** Baseline characteristics, laboratory, and clinical finding of study patients.

Isolated Fallopian Tube Torsion	
All Patients	*n* (%)
**Baseline Characteristics**	
Age, mean ± SD, years	14.5 ± 1.43
**Laboratory Findings**	
WBC, mg/dL	10.47 ± 2.40
Neutrophils, %	76.9 ± 7.58
Lymphocytes, %	18.1 ± 5.94
BUN, mg/dL	16.24 ± 8.19
Crea, mg/dL	0.73 ± 0.24
ALT, mg/dL	17.21 ± 7.23
AST, mg/dL	19.44 ± 7.31
ALP, mg/dL	86.24 ± 48.51
BS, mg/dL	94.11 ± 18.82
**Clinical Findings**	
Presence of pain	10 (100)
Time to start pain	
Less than 24h	7 (70)
More than 24h	3 (30)
Shape of pain	
Colic	3 (30)
Continuous pain	7 (70)
Nausea	5 (50)
Vomiting	5 (50)
Vaginal discharge	0 (0)
Menarche	9 (90)
Physical examination	
Abdominal tenderness	9 (90)
Abdominal defender	5 (50)
Abdominal rebound	3 (30)
Imaging methods	
Direct abdominal radiography	10 (100)
US	10 (100)
MRI	2 (20)
Tumor Markers	2 (20)
Paraovarian cyst	7 (70)
Doppler current reduction/absence	3 (30)
Surgical intervention method	
Open surgery	8 (80)
Laparoscopic	2 (20)
Place of torsion	
Right	5 (50)
Left	5 (50)
Operation performed	
Detorsion	5 (50)
Salpingectomy	5 (50)

SD: standard deviation, WBC: white blood cell, BUN: blood urea nitrogen, Crea: creatinin, ALT: alanine aminotransferase, AST: aspartate aminotransferase, ALP: alkaline phosphatase, BS: blood sugar, US: ultrasonography, MRI: magnetic resonance imaging.

**Table 2 diagnostics-09-00110-t002:** Characteristics of fallopian tube torsion cases.

Age	Pain Localization	RDUS	MRI	Preop Diagnosis	Postop Diagnosis	Accompanying Pathology	Procedure
14	Lower left	Left paraovarian cyst, free liquid	None	Acute abdomen	Left IFTT	None	Open salpingectomy
13	Lower right	Normal	None	Acute abdomen	Right IFTT	None	Open detorsion, cyst excision
14	Lower left	Right paraovarian mass	Semi-solid cystic paraadnexial	Paraovarian mass	Right IFTT	Paratubal cyst	Open salpingectomy, cyst excision
15	All abdomen	Right paraovarian cyst, free liquid	None	Right ovarian torsion	Right IFTT	Paratubal cyst	Open salpingectomy, cyst excision
17	Lower left	Left paraovarian cyst	None	Left ovarian torsion, left IFTT	Left IFTT	None	LAP detorsion
16	Lower right	Right paraovarian cyst, free liquid	None	Right ovarian torsion	Right IFTT	None	LAPdetorsion
12	All abdomen	Normal	Fallopian tube wall thickening, hydrohematosalpinx	IFTT	Right IFTT	None	Opensalpingectomy
15	Lower left	Free liquid	None	Ovarian cyst rupture	Left IFTT	Paratubal cyst	Open detorsion, cyst excision
14	Sub-dial	Left paraovarian cyst	None	Acute abdomen	Left IFTT	Paratubal cyst	
15	Lower left	Left paraovarian cyst	None	Left ovarian torsion	Left IFTT	None	Open detorsion

RDUS: color Doppler ultrasonography, MRI: magnetic resonance imaging, IFTT: isolated fallopian tube torsion LAP: lymphadenopathy.

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
