# Peer review of "A Fifteen-Year Analysis of Rare Isolated Fallopian Tube Torsions in Adolescent Children: A Case Series"

_diagnostics, 2019, doi:10.3390/diagnostics9030110_

Round 1

Reviewer 1 Report

The authors present an interesting work that explains the diagnostic, preoperative and therapeutic management of adolescents who have Isolated tubal torsion.
Acute abdominal pain in adolescents is a clinical condition that can have a significant impact on fertility, and the right diagnostic and therapeutic management can be crucial.
It is important to add a paragraph in the discussion regarding the importance and usefulness of laparoscopy. Particularly in very young patients, the use of single port surgery can be considered useful and effective to confirm the diagnosis and achieve adequate treatment. In the literature there are some works that would be useful to cite (Perioperative Outcomes of Laparoendoscopic Single-Site Surgery (LESS) Versus Conventional Laparoscopy for Adnexal Disease: A Case-Control Study. Surgical Innovation Volume 18, Issue 1, March 2011, Pages 29-33 ) (Laparoendoscopic Single-site Surgery for the Treatment of Benign Adnexal Disease: A Prospective Diagnostic and Therapeutic Endoscopy Open Access 2010, Article number 108258)
(Laparoendoscopic single-site surgery for the treatment of benign adnexal diseases: A pilot study
Surgical Endoscopy Volume 25, Issue 4, April 2011, Pages 1215-1221)

Author Response

Dear Reviewer,

1- English language and style have been revised.

2- The conclusions supported the results but were further strengthened.

3- A paragraph was added to the discussion about the importance and usefulness of laparoscopy. The discussion was further strengthened.

Reviewer 2 Report

Isolated fallopian tube torsion (IFTT) is an important clinical entity due to low alertness of this diagnosis in the differential diagnosis of lower abdominal pain in childhood

The authors present 10 patients presenting to the emergency room with abdominal pain diagnosed at the operation as  IFFT.

The introduction is nearly totally (only row 36)deprived of data regarding ultrasound. One of the largest series reported so far regarding IFTT  focusing on the sonographic diagnosis of IFTT was published in 2017 ( Raban et al). Would recommend to discuss it in the introduction and  also discussion section related to your results. In the conclusion of your article you mention the importance of US albeit in the discussion section. In 4 patients torsion of ovary was suspected - on what sonographic basis - can please detail since after all at the end this diagnosis of torsion although not of the tube, did direct to OR which is the needed treatment .  8/10 patients were operated by laparotomy and not laparoscopy. Any reason for that?

Author Response

Dear Reviewer,

1- English language and style have been revised.

2-The entrance provided a sufficient infrastructure and, although it contained all relevant references, was still strengthened by revision.

3- The literature on ultrasound was included in both the introduction and discussion section.

Round 2

Reviewer 1 Report

The authors showed that Isolated tubal torsion should be considered in children presenting with acute abdominal pain in early adolescence. Early diagnosis is important for the maintenance of fertility.

Reviewer 2 Report

No remarks

Round 3

Reviewer 2 Report

Accept